# The Effects of a Bacterial Endotoxin on Behavior and Sensory-CNS-Motor Circuits in *Drosophila melanogaster*

**DOI:** 10.3390/insects10040115

**Published:** 2019-04-22

**Authors:** Oscar Istas, Abigail Greenhalgh, Robin Cooper

**Affiliations:** Department of Biology, University of Kentucky, Lexington, KY 40506-0225, USA; owis222@g.uky.edu (O.I.); Abigail.Greenhalgh@uky.edu (A.G.)

**Keywords:** lipopolysaccharides, peptidoglycans, bacteria, synapse, integration, sensory, motor, Drosophila

## Abstract

The effect of bacterial sepsis on animal behavior and physiology is complex due to direct and indirect actions. The most common form of bacterial sepsis in humans is from gram-negative bacterial strains. The endotoxin (lipopolysaccharide, LPS) and/or associated peptidoglycans from the bacteria are the key agents to induce an immune response, which then produces a cascade of immunological consequences. However, there are direct actions of LPS and associated peptidoglycans on cells which are commonly overlooked. This study showed behavioral and neural changes in larval *Drosophila* fed commercially obtained LPS from *Serratia marcescens*. Locomotor behavior was not altered, but feeding behavior increased and responses to sensory tactile stimuli were decreased. In driving a sensory-central nervous system (CNS)-motor neural circuit in in-situ preparations, direct application of commercially obtained LPS initially increased evoked activity and then decreased and even stopped evoked responses in a dose-dependent manner. With acute LPS and associated peptidoglycans exposure (10 min), the depressed neural responses recovered within a few minutes after removal of LPS. Commercially obtained LPS induces a transitory hyperpolarization of the body wall muscles within seconds of exposure and alters activity within the CNS circuit. Thus, LPS and/or associated peptidoglycans have direct effects on body wall muscle without a secondary immune response.

## 1. Introduction

Septicemia is estimated to account for 17% of the death rate in people hospitalized in the USA [1]. In the case of bacterial septicemia, many of the cases are due to gram-negative forms. These gram-negative strains release the endotoxin lipopolysaccharides (LPS) from the outer layer of the membrane. LPS is responsible for activating the CD14/TLR4/MD2 receptor complex [2,3] and can trigger an immune response resulting in the release of cytokines and nitric oxide from immune cells as well as various tissues in the body, including cells in the brain [4,5]. LPS has indirect effects on mammalian neurons starting with activating the Toll receptor complex (CD14/TLR4/MD2); however, in *Drosophila melanogaster* where the Toll receptor was discovered [6,7], it does not appear LPS mediates its response through the Toll receptor complex [8,9]. Rather, in *D. melanogaster* the Immune deficiency (Imd) signaling pathway is the main cellular cascade stimulated by associated peptidoglycans from gram-negative bacteria and not LPS itself [8,10,11]. However, the expression profiles for peptidoglycans receptors in the brains of insects have yet to be fully identified nor the effects on the physiology of the central neural circuits. *D. melanogaster* avoid eating food containing bacterial LPS, which likely also contains the associated peptidoglycans [12,13]. This gustatory avoidance was shown to be mediated through a TRPA1 receptor [12]. It is important to note that commercially obtained LPS likely contains associated peptidoglycans from the same strain of gram-negative bacteria [8]. Thus, exposed preparations reported on in the past or present with commercially obtained LPS is likely a mixture, but still represents the effects of what compounds the tissue or animal would be exposed to from gram-negative bacteria. 

In the mammalian brain, it is known that exposure to LPS can increase intracellular Ca^2+^ in some cell types which results in apotosis [14]. The cellular interactions with microglia and various cell types are still being elucidated. Recent studies have demonstrated that glutamate is released by exposure to LPS and excites neighboring neurons [13]. Direct cellular response from the action of LPS on glutamate receptors has not yet been established. However, activation of TLR4 receptors by LPS results in an interaction of TLR4 receptors and the N-methyl-D-aspartate (NMDA) glutamate receptor in microglial cells [15,16].

At the motor nerve terminals of crayfish, LPS (1–2 µg/mL LPS and potentially associated peptidoglycans from *Serratia marcescens*) increases evoked release and increases the frequency of spontaneous miniature synaptic potentials [17]. LPS and peptidoglycans (10–50 µg/mL, *Salmonella typhimunum*) exposure at the frog neuromuscular junction (NMJ), which is cholinergic, also increases the frequency of spontaneous miniature synaptic potentials but blocks evoked release [18]. The reduction of evoked release at the frog NMJ was suspected to be due to blocking the pre-synaptic voltage-gated calcium channels, but no direct measures were obtained. This phenomenon also occurs at frog NMJs with LPS and peptidoglycans (10–50 µg/mL) from *Escherichia coli* [19]. Thus, no evoked synchronized release occurred with electrical stimulation of the motor nerve. There does not appear to be a consistent response on direct actions of LPS on evoked synaptic transmission. The responses at frog NMJs were not able to recover after washing away the LPS and peptidoglycans but the enhanced evoked responses at the crayfish NMJ were able to be reversed by removing LPS containing saline. In the hippocampal slice of a rodent brain, the depressing effects were able to be reversed by flushing away the LPS and peptidoglycans and was postulated to be due to blocking Ca^2+^ entry through NMDA receptors [20]. The depressing effects on central circuits were assumed to be the cause of amnesic actions of bacterial infection in neural tissue [20].

The larval *D. melanogaster* serves as a model for investigating mechanistic properties of synaptic transmission in a neural circuit by activating sensory neurons and monitoring motor neuron activity. The sensory-CNS-motor neural circuit was demonstrated previously to be a good model to address the effects of modulators (i.e., serotonin, dopamine, octopamine) affecting larval behaviors [21,22]. Since it was recently shown that LPS and the associated peptidoglycans from *Serratia marcescens* (*S. m.*) could acutely enhance the heart rate and then depress it over a few minutes in in-situ preparations, bathed in saline, of larval *D. melanogaster* as well as depresses evoked synaptic transmission at the larval neuromuscular junction (NMJ) [23,24], we hypothesized that sensory perception would also likely be altered within the central synapses in the CNS. In our in-situ approach, the sensory receptors are by-passed as the nerves are electrically stimulated to directly activate them in a sensory-CNS-motor nerve circuit. Since it was noted that synaptic transmission is compromised at the NMJ in larval *D. melanogaster* [24], we also expected larval locomotion to be retarded if they larvae were to consume LPS and the associated peptidoglycans at high concentrations which may overwhelm the innate immune response and potentially systemically infiltrate the larvae with the compounds without having to inject the larvae.

In this study, we address the acute effects of LPS and associated peptidoglycans from commercially obtained *Serratia marcescens* (*S. m.*) on the physiological function of an activated sensory-CNS-motor nerve circuit as well as the effects of feeding these bacterial components to developing larvae. 

## 2. Materials and Methods

Wild type Canton S (CS) *D. melanogaster* were used. This strain has been isogenic in the lab for more than 10 years and was originally obtained from Bloomington Stock Center. To obtain staged larvae, the flies were held at 25 °C in a 12 h light/dark incubator before being tested. All animals were maintained in vials partially filled with a cornmeal-agar-dextrose-yeast medium.

### 2.1. Behaviors

Early 3rd instar locomotive behavior was evaluated as described in Neckameyer [25] and Li et al. [26]. In brief, single animals were moved to an apple-juice agar (1% agar) surface following exposure to a controlled concentration of LPS in tainted food for 48 h. The LPS is mixed in standard cornmeal food at the desired concentration. The dilution used was 1 g of moist food was equivalent to 1 mL in volume. The LD_50_ in rodents for injected LPS from *S. m.* is 650 µg/mL [14] (6 × 10^6^ CFU- colony-forming units, Iwaya et al. [27]). Thus, as for native rodents, *D. melanogaster* are also exposed to high levels of gram-negative bacterial strains in their native environment. For this reason, a high concentration of 500 µg/ml was used in this study for *D. melanogaster* over the feeding of 48 h. Note that the larvae are not injected with bacteria but fed the commercial LPS and associated peptidoglycans. The number of body wall contractions, quantified by recording posterior to anterior peristaltic contractions, was counted for 1 min under dim lighting in room temperature (22–23 °C). The feeding behavior assessment was conducted by placing larvae in shallow water and yeast and visually counting the mouth hook movements. A Appendix A as a video is provided of this behavior (Appendix A) as well as a YouTube link (https://youtu.be/0VJx6bYpruc). In this condition *D. melanogaster* larvae immediately feed, initiating a pattern of repetitive mouth hooks movements that allow for food intake. This method of observing the larvae in dilute food also stabilizes the larvae, making it easier to observe mouth hook extensions and contractions.

The mechanosensory behavioral assay was as previously described [28]. The slight touch was placed on the lateral side of the larvae while they were crawling on the apple juice agar plates. 

### 2.2. Electrophysiology

The technique to dissect larvae and investigate the function of the sensory-CNS-motor nerve circuit is described in Dasari and Cooper [21]. In brief, a longitudinal dorsal midline cut was made in 3rd instar CS larvae to expose the CNS. Two of the last segmental nerves were cut and sucked into a suction electrode, which is filled with HL-3 saline and stimulated. The segmental roots were severed from the body wall to selectively stimulate sensory nerves orthodromically. The segmental nerves were stimulated with trains of pulses, the paradigm maintained at 10–20 pulses per train at 40–60 Hz (S88 Stimulator, Astro-Med, Inc., Grass Co., West Warwick, RI, USA). The modified HL3 saline was used for physiological measures [29] at a pH of 7.1 [30]. Saline solution (in mM): 1.0 CaCl_2_·2H_2_O, 70 NaCl, 20 MgCl_2_, 5 KCl, 10 NaHCO_3_, 5 trehalose, 115 sucrose, 25 5N, N-bis(2-hydoxyethyl)-2-aminoethanesulfonic acid (BES). 

There was a 10 s delay from one stimulation train to the next stimulation train. The voltage was dependent on the initial observation of evoked responses, and generally varied between 4–10 volts because the suction electrodes, which were used to stimulate the segmental nerves in each preparation, were slightly different. Synaptic responses at the larval *D. melanogaster* NMJs were recorded by standard procedures [31]. Thus, segmental nerves were stimulated with a controlled frequency and voltage until a response was observed from an intracellular microelectrode in muscle fiber 6 (m6) contralateral (across the midline) to the stimulus. This allows for the examination of activity within the CNS associated with a controlled afferent nerve stimulus and the associated motor output. The traces were measured by averaging the EJP frequency in 5 stimulus trains made with normal saline and 5 stimulus trains after exchanging saline with LPS containing saline after 10 min of exposure. Individual trains of pulses elicit bursts of EJPs that were quantified through manual counting (see Figure 1). To ensure preparation viability following the application of each compound, the compounds were washed out and replaced with normal saline. The average frequency of EJPs from each animal and the means from each treatment group were compared. This concentrations of LPS used were to allow for comparison to previous studies [23,24]. Data were recorded as percent change from a saline solution to a saline solution containing the compound of varying concentration in order to generate a dose-response relationship.

All experiments were performed at room temperatures (20–21 ℃). The excitatory junction potentials (EJPs) were measured by intracellular recordings with a sharp glass electrode (3 M KCl) and AxoClamp-2 B amplifier (Molecular Devices, LLC. 1311 Orleans Drive, Sunnyvale, CA, USA). Preparations were used immediately after dissection. Electrical signals were recorded online to a computer via a PowerLab/4 s interface (ADI Instruments, Colorado Springs, CO, USA).

Commercial LPS from *Serratia marcescens* (*S. m.*) was dissolved in physiological saline the day of experimentation. This LPS may also contain some associated peptidoglycans from *S. m.* All chemicals were obtained from Sigma-Aldrich (St. Louis, MO, USA). All electrophysiological measures were made in muscle 6 of segments 3 or 4. 

Data are expressed as an average value along with the standard error of the mean (i.e., ±SEM) or as a percent change. The rank sum pairwise tests or a sign test was used to compare the differences in responses before and after exchanging solutions. In some cases, synaptic responses were non-existent with exposure to LPS which did not allow parametric analysis. The analysis was performed with Sigma Stat software. *p* of ≤0.05 is considered as statistically significant. 

## 3. Results

Providing food tainted with LPS was an approach to determine if there were effects on behaviors which would mimic alteration associated with neural activity by direct application of LPS to the CNS.

### 3.1. Impact of Oral Supplementation of LPS on Larval Locomotion and Feeding

After 48 h of being exposed and eating LPS-tainted food, the behavioral analysis revealed an increase in the larvae mouth hook movement (N = 20, T-test *p* < 0.05) without any significant changes in body wall movements as compared to controls (Figure 2; N = 20, T-test *p* > 0.05). The rationale for the long-term feeding assay was determined if the LPS and associated peptidoglycans may breach the gastrointestinal tract and drastically reduce these behaviors since it is established that direct application with in-situ preparations depresses synaptic transmission at NMJs [24]. To determine if of LPS at 500 µg/mL was causing the larvae to be too compromised for behavioral analysis, two higher concentrations were used to assess survival. Food tainted to 750 µg/mL and 1000 µg/mL of LPS for 48 h resulted in one larva out of 22 dying in the food at 1000 µg/mL. Also, one larva out of 22 pupated and appeared dead as a pupa in the 750 µg/mL exposure for 48 hrs. All the larvae in both high concentrations were eating and crawling. Thus, the larvae used for the behavioral assays at 500 µg/mL were not adversely affected. An LD50 was not performed by feeding LPS, but obviously, the larvae can survive in high concentrations of LPS in the diet.

Even though larvae that consumed LPS did not demonstrate a significant change in locomotion, as measured by contraction of the body wall movements in an inchworm manner, there appeared to be some observable differences with respect to the mechanosensory touch of the larvae. Thus, measures were undertaken to determine if there were responsive differences to a slight mechanosensory touch on the lateral side of crawling larvae [24]. The behaviors were devised into an ethogram based on the observed responses. Results revealed a decrease in responsiveness in LPS-fed larvae (Figure 3A, N = 20). In sub-dividing the behavioral repertoire into the observed behaviors, the slight bend and strong bend toward the stimulus were the responses most notably decreased in LPS fed larvae (Figure 3B). The response to the back up (inchworm backwards) was increased in the LPS fed larvae (Figure 3B). An ethogram with behavioral positions is provided as a Appendix A along with the raw data from this assay (Appendix A).

### 3.2. LPS Modulation of Sensorimotor Circuit Activity

The direct application of LPS and associated peptidoglycans to the exposed CNS during evoked stimulation revealed interesting results. With the 500 µg/mL exposure, the evoked bursts became irregular by firing immediately, where in some cases there was an increase in the duration and others a decrease in the burst duration (Figure 4A). The bursts of evoked EJPs were relatively consistent when bathed in saline under a given stimulation paradigm (Figure 4B). After exposure to LPS/peptidoglycans, the bathing saline containing this mixture was removed and the bath was exchanged at least 3 times with fresh saline not containing the mixture. Evoked responses returned in 9 of 10 preparations for the 500 µg/mL trials after removing the mixture. Recovery would occur within 1 or 2 min following the saline exchanges (Figure 4C). After 5 min and 10 min of exposure, while stimulating the sensory nerves, the frequency of activity by the motor nerves decreased substantially for 500 µg/mL (N = 10, Sign test *p* < 0.05). However, for 100 µg/mL the activity was mixed. Within the first 5 min of exposure to 100 µg/mL LPS 9 of 11 preparations increased activity in a bursting fashion for a few minutes (Sign test *p* < 0.05). By 10 min of exposure to 100 µg/mL LPS 9 of the 11 preparations decreased activity (Sign test *p* < 0.05) with some showing no inducible activity, and 10 of the 11 preparations showed a decrease in the EJP amplitude (Sign test *p* < 0.05). The compiled data for 100 µg/mL and 500 µg/mL for altering motor neuronal drive is presented as a percent change from pre-exposure to exposure of LPS for the responses when a maximal change was recorded (Figure 4D). One preparation with 100 µg/mL presented with a low frequency of activity in saline and a much higher evoked frequency after being incubated for 10 min in LPS presenting an outlier in the analysis of percent change in frequency. Thus, a mean (±SEM) percent change in frequency is presented with (−36.9, ±45) and without (−81, ±10) the outlier for 100 µg/mL (Figure 4D). The 500 µg/mL presented a mean decrease of 59 (±15). The percent changes for individual preparations are depicted by the closed circles next to the mean (±SEM) bar graphs. 

Exposing the body wall muscles and neuromuscular junctions to LPS revealed a novel phenomenon. The muscle showed a hyperpolarization of the membrane potential (Figure 4A). The membrane potential would rapidly hyperpolarize followed by gradually returning to basal levels within 5–10 min. Upon washing away LPS containing saline the membrane potential was close to the initial values. However, some muscle fibers became depolarized over 10 min of incubation with LPS and exchange of the bathing media back to standard saline. This depolarization is likely due to some damage of the muscle during the evoked responses while maintaining an intracellular electrode across the twitching muscle membrane.

## 4. Discussion

Tainting food with LPS mixture induced a behavioral change, but it is unknown what concentration of LPS or associated peptidoglycans are present, if any at all made it to circulating levels within the hemolymph. It is odd that the circuit regulating mouth hook movements was enhanced while the response to tactile sensitivity was reduced when the larvae were crawling. Since adult *Drosophila* avoid eating food containing bacterial LPS and associated peptidoglycans [12,13], it might be that the larvae reduce their feeding but when given the yeast solution in water for the behavioral assay speed up their food intake. However, the larvae did not appear to be any smaller in the ones fed tainted food as compared to the controls eating a standard cornmeal diet. Perhaps longer exposure times or earlier developmental stages, due to enhanced sensitivity, would have different consequences. It is unknown how the gastrointestinal tract and the associated intrinsic bacteria in the intestines might respond during the period of LPS exposure. It is possible the innate immune system may be enhanced throughout the time of exposure. Sickness due to infection (bacterial or viral) induces a decrease in appetite in mammals and is assumed to be beneficial for fighting the infection to direct energy resources to the immune system of the host [32,33]. However, crickets (*Gryllus texensis*) injected with heat-killed bacteria containing LPS and associated peptidoglycans (*S. marcecens*) did not show reduced feeding or locomotion [34]. As suggested by Sullivan et al. [34], ectotherms and endotherms may show differences in behaviors associated with sickness, likely due to the lack of pyrogenic effects in some insects.

The direct exposure to 100 µg/mL of the LPS/associated peptidoglycans initially promoted the recruiting of motor neurons by the sensory drive; however, it is unknown if the presynaptic endings of sensory neurons, receptivity of dendrites for interneurons, or a combination of potential factors might be a target of action by LPS/associated peptidoglycans. The sensory neurons are cholinergic and it is known that muscarinic as well as nicotinic receptors reside in the CNS [35,36,37,38]. At the cholinergic neuromuscular junction of the frog, there is no indication of increased sensitivity to the neurotransmitter during LPS/associated peptidoglycans exposure and it was shown that the receptors remained functional due to the presence of increased occurrences of spontaneous quantal responses while the nerve terminal was slowly unable to provide evoked responses [18,19]. Thus, it seems unlikely that the receptivity to acetylcholine (Ach) is compromised. Glutamatergic receptivity at the crayfish neuromuscular junction, which is similar in the receptor subtypes on *Drosophila* body wall muscle [39], was not compromised by LPS/associated peptidoglycans (*S. m.*) exposure. Instead, the evoked release was enhanced and was suggested to be caused by promoting presynaptic Ca^2+^ influx with the presynaptic terminals due in part by the increased occurrences of spontaneous quantal responses [17]. Given the complexity of neuromodulatory responses known to occur within the larvae brain by dopamine, serotonin, octopamine, Ach, and GABA [22,40,41,42,43,44], there is a myriad of possibilities to investigate in the site(s) of action by LPS/associated peptidoglycans influencing the driven sensory-CNS-motor circuit. 

The mix of excitatory and depressing CNS effects after LPS/associated peptidoglycans exposure is not unique to the larval *D. melanogaster* brain. LPS (*Escherichia coli*) exposure has been reported to facilitate epileptiform activity in vitro [45], while another study of the rat hippocampal CA1 area indicated LPS suppressed NMDA receptor-mediated excitatory postsynaptic currents, potentially by blocking Ca^2+^ entry through NMDA receptors relating to the amnesic action of bacterial sepsis [20]. The presence of microglia and astrocytes, which can be activated by LPS and maybe also by associated peptidoglycans, offers another level of complexity in the intact CNS of mammals. The action of LPS on microglia was shown to release proinflammatory cytokines (IL-1β and TNF-α) which led to enhanced glutamate release and a subsequent reduction in the excitatory synaptic activity of pyramidal neurons in rodent hippocampal slices [46]. Also, in some conditions, LPS exposure enhanced long-term depression in hippocampal slices via activation of microglia [47]. In comparison to the findings in the larval *D. melanogaster* CNS, the depressed neural responses are generally reversible with an acute 10 min of LPS/associated peptidoglycans exposure. The mechanisms to account for the depressed effect in the larval brain does appear to be concentration dependent, as the high concentration leads to a more rapid depression of the sensory driven response than the lower concentration. 

The rapid hyperpolarization of the body wall muscle with the high LPS exposure occurs by a yet unknown mechanism. We are currently addressing the potential mechanism in a subsequent more in-depth study on the effects of LPS directly on the body wall muscles of *D. melanogaster* larvae. The effect does not appear to be due to calcium-activated potassium channels [48] or activated nitric oxide synthase (NOS) as indicated in the preliminary studies underway [24]. Future studies need to address if these reported responses are due to purified LPS or the associated peptidoglycans [8] as then one may be able to better addresses which receptors and the binding sites on the three known PGRPs in *D. melanogaster* (PGRP-SA, PGRP-LC, and PGRP-LE). It is known PGRP-LC and PGRP-LE respond to constituents of gram-negative bacteria [49,50]. The location and function of these receptors in various tissues of larval *D. melanogaster* has not yet been fully characterized. The ability to genetically reduce or enhance LPS receptors (Imd and/or Toll) and their associated cellular cascade in defined neurons [11] will allow one to further dissect the acute and long-term effects on the defined neural circuits. Perhaps further investigations using the genetically amenable *D. melanogaster* model and the ability to express Ca^2+^ indicators in defined neurons within known neural circuits and neuromuscular junctions will shed light on the mechanism of diverse action, not only applicable in this model animal, but also in mammalian systems.

## 5. Conclusions

The findings indicate that feeding larvae commercial LPS mixture from *S. m.* for 48 h reduced their sensitivity to mechanosensory stimulation. However, feeding induced an increase in mouth hook movements, indicating enhanced feeding behavior. No alteration in locomotive behavior was observed. In-situ preparations, with direct application of LPS mixture to the CNS, initially increased the frequency in the sensory drive of motor nerve activity at 100 µg/ml followed by a substantial decrease within 10 min to being unable to stimulate the sensory-CNS-motor circuit. Exposure to 500 µg/mL would initially cause a few bursts of activity followed by rapidly decreasing motor nerve activity. In addition, the high concentration of LPS mixture decreased the EJP amplitude and produced a transient hyperpolarization of the body wall muscle. In the majority of preparations, the depressing effect of LPS on the sensory-CNS-motor nerve circuit was able to be partly reversed by exchanging the bathing media with fresh saline.

## Figures and Tables

**Figure 1 insects-10-00115-f001:**
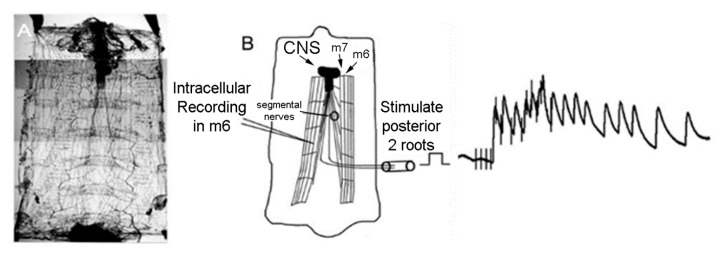
Schematic diagram of the *D. melanogaster* larva preparation for activating and recording a sensory-CNS-motor nerve circuit. (**A**) The preparation is pinned at the four corners to keep the preparation taut. The ventral abdominal muscle, m6, was used in this study. The ventral abdominal muscle m7 is shown as reference. (**B**) The segmental nerves were stimulated by placing the cut nerve roots into the lumen of a suction electrode and recruiting various subsets of sensory neurons. A sample of evoked response recorded with an intracellular electrode in m6 of EJPs, from a stimulus train, is shown.

**Figure 2 insects-10-00115-f002:**
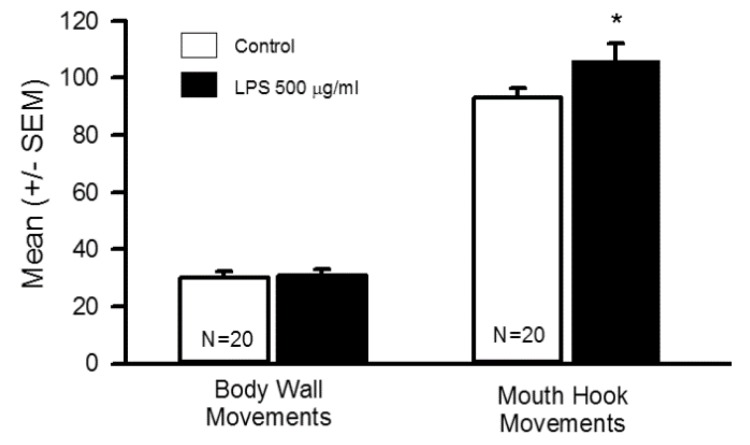
The results of the larval behavioral assays after consuming LPS for 48 h. The larvae mouth hook movements were significantly increased as compared to controls (N = 20, T-test *p* < 0.05).

**Figure 3 insects-10-00115-f003:**
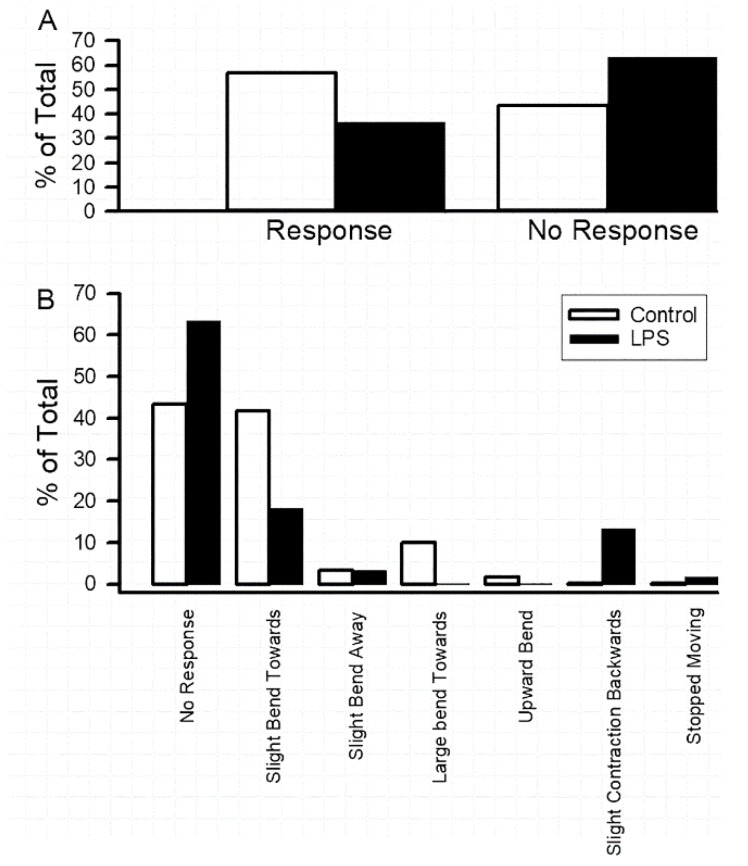
Responses to light mechanosensory touch. The behaviors were devised into an ethogram based on the observed responses. Examining if there was a response or not to the stimulus revealed a decrease in responsiveness in LPS-fed larvae (**A**) (N = 20). In sub-dividing the behavioral repertoire into the observed behaviors, the slight bend and strong bend toward the stimulus were the responses most notably decreased in the LPS-fed larvae (**B**). The response to the back up (i.e., inchworm backwards) was increased in the LPS-fed larvae when touched (**B**). The other observed behaviors did not occur often enough for strong observational differences from the control larvae. (Appendix A).

**Figure 4 insects-10-00115-f004:**
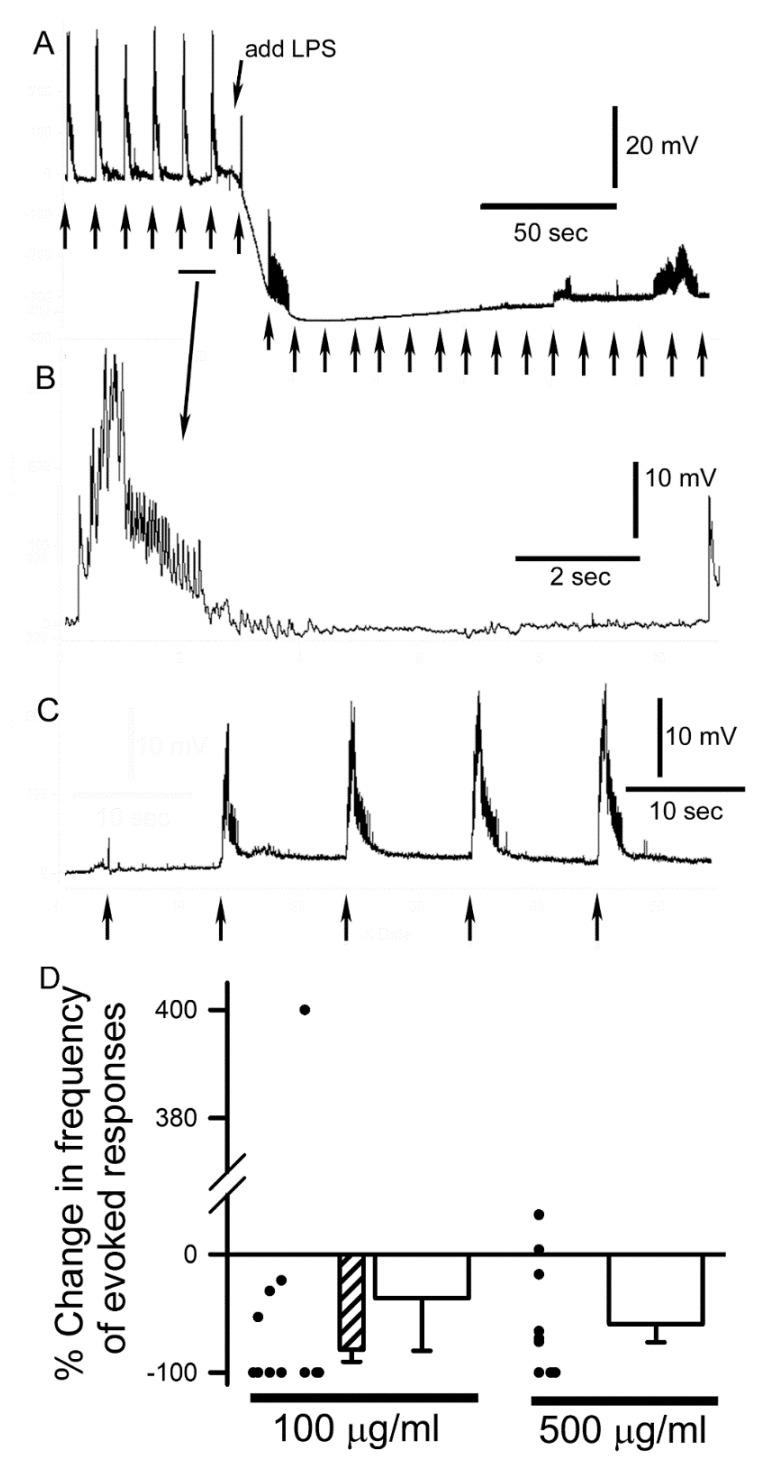
The effect of LPS exposure on the evoked sensory-CNS-motor neural circuit and synaptic transmission at the neuromuscular junction in 3rd instar larvae. (**A**) The stimulus train delivered to sensory roots every 10 s before and during LPS exposure evoked a motor nerve recruitment which was indicated by the evoked EJPs measured in muscle 6. Note the enlarged trace shown in (**B**) of the 5th–6th stimulus train in A above. The muscle membrane potential rapidly hyperpolarized upon exposure to LPS and the amplitude of the EJPs also decreased rapidly. (**C**) Replacing the saline containing LPS with fresh untainted saline resulted in the evoked EJPs to return and the amplitude of the EJPs to increase as compared to during the LPS exposure. (**D**) Average percent change in EJP frequency in response to 10 min of LPS exposure for 100 µg/mL and 500 µg/mL. The mean (±SEM) percent changes are shown as bars along with the changes in the individual preparations. The individual preparations are offset from each other in order to prevent overlap. The outlier at 400% increase for the 100 µg/mL was removed for additional analysis of a mean (±SEM) as seen in the hatched bar graph.

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
