# Peer review of "The Effects of a Bacterial Endotoxin on Behavior and Sensory-CNS-Motor Circuits in Drosophila melanogaster"

_insects, 2019, doi:10.3390/insects10040115_

Round 1

Reviewer 1 Report

In this manuscript, Oscar et al showed that feeding of LPS/peptidoglycans could affect the body wall muscle of Drosophila larvae, and that it might reduce their sensitivity to mechanosensory stimulation. Their study is intriguing in terms of novel examination of the effect of immuno-stimulatory molecules on the nervous system, and it could be suitable for publication in the Journal Insects. However, it seems to the reviewer that some concerns could be addressed in order to strengthen the manuscript. 

Concerns:

1/ The reviewer has concern about reproducibility of some results in this paper, since it seems to be missing that how many times they perform those experiments. If possible, the reviewer would like to see the repeated data they do not use in this manuscript, as for "reviewer only".

2/ Line 22-23 in abstract, they write "..... without a secondary immune response.", but the reviewer is not able to find experimental evidence in the manuscript that supports this statement.

3/ As they clearly recognized that "commercial LPS mixture" is contaminated with peptidoglycans. Thus, they should consistently use the term "LPS and associated peptidoglycans", even in the title of this paper.

Reviewer 2 Report

Title: The effects of a bacterial endotoxin (LPS) on behavior and sensory-CNS-motor circuits

Overview: The authors conducted a behavioral and physiological study on the effects of bacterial lipopolysaccharide using Drosophila as their model system. Using Drosophila makes it a model broadly acceptable by a broader scientific community and thus increasing the impact of this study. However, the authors need to address some major concerns regarding the manuscript.

Overall comments: Authors need to determine the LD50s of the endotoxin before testing the behavioral responses. Also, the concentrations used in the study need to be justified. Methods needs more details in terms of number of larvae used. Raw data of ethogram need to be provided as a supplementary file. A video recording of the larval feeding behavior will be helpful in justifying the results observed in this study.

Specific comments:

Line 52: Expand the abbreviation NMDA

Line 56: Expand the abbreviation NMJ

Line 132: Move the composition of HL3 buffer and preparation of LPS working stock to the first paragraph of section 2.2.

Line 151: Change determine to determined

Line 166: Are these responses to movements? Response is a very broad term, this need to be specific to the data recorded.

Line 177: Replace “some” with specific percent of larvae that showed this response.

Line 193: How much reduction in response is observed after 5-10 minutes of exposure to LPS?

Line 215: Wrong choice of word – “intact”

Line 227: “low concentrations” – Need to mention specific concentration

Reviewer 3 Report

The article "the effects of a bacterial endotoxin (LPS) on behavior and sensory-CNS-motor circuits" by Istas et al, discusses if LPS can affect the behavioral and sensory reactions in Drosophila. The article presents some interesting results and I recommend the publication of the article.

Round 2

Reviewer 2 Report

Authors have done a significant improvement to the manuscript and addressed all the concerns. I recommend the manuscript for publication.

Author Response

The manuscript has been positively evaluated by three referees and by myself. However, prior acceptance for publication in Insects, a few minor revisions are still needed.

1. All over the text (title, abstract, intro, material and methods and discussion) authors should make some efforts to mention the species name. Drosophila melanogaster should be provided in title and in most parts of the manuscript where authors refer to   Drosophila (including figure caption), animals, larvae, wild type, etc.

Response: Done. Drosophila melanogaster is now used throughout. Also, that the wild type Canton S was used.

2. Moreover, the proposed title looks odd and I suggest removing ‘’(LPS and 2 associated peptidoglycans)”

Response: The revised title is:

The effects of a bacterial endotoxin on behavior and sensory-CNS-motor circuits in Drosophila melanogaster

Insects EISSN 2075-4450 Published by MDPI AG, Basel, Switzerland RSS E-Mail Table of Contents Alert
Back to Top